# Neuroethology of the Waggle Dance: How Followers Interact with the Waggle Dancer and Detect Spatial Information

**DOI:** 10.3390/insects10100336

**Published:** 2019-10-11

**Authors:** Hiroyuki Ai, Ryuichi Okada, Midori Sakura, Thomas Wachtler, Hidetoshi Ikeno

**Affiliations:** 1Department of Earth System Science, Fukuoka University, Fukuoka 814-0180, Japan; 2Department of Biology, Kobe University, Kobe 657-8501, Japan; 3Department of Biology II, Ludwig-Maximilians-Universität München, Planegg-Martinsried 82152, Germany; 4Department of Human Science and Environment, University Hyogo, Kobe 670-0092, Japan

**Keywords:** honeybee, waggle dance, distance information, brain, antenna-mechanosensory center, vibration, sensory processing, standard brain, computational analysis, polarized light processing

## Abstract

Since the honeybee possesses eusociality, advanced learning, memory ability, and information sharing through the use of various pheromones and sophisticated symbol communication (i.e., the “waggle dance”), this remarkable social animal has been one of the model symbolic animals for biological studies, animal ecology, ethology, and neuroethology. Karl von Frisch discovered the meanings of the waggle dance and called the communication a “dance language.” Subsequent to this discovery, it has been extensively studied how effectively recruits translate the code in the dance to reach the advertised destination and how the waggle dance information conflicts with the information based on their own foraging experience. The dance followers, mostly foragers, detect and interact with the waggle dancer, and are finally recruited to the food source. In this review, we summarize the current state of knowledge on the neural processing underlying this fascinating behavior.

## 1. Behavioral Significance of the Waggle Dance

The honeybee (*Apis mellifera*) is well known to possess the ability to communicate to its nestmates to convey the locational information of a profitable food source she has visited [1]. Once a scout bee has found a profitable food source and returned to the hive, she will perform a dance (round dance or waggle dance for a short or long distance food source, respectively) to recruit new bees to visit the food source (Figure 1). During the waggle dance, the bee runs relatively straight with her body waggling (a waggle run) and circles back to the starting point of the waggle run without wagging or wing-beating (the return run). On the vertical comb, the direction to a food source from the hive relative to the sun’s azimuth is encoded according to the angle between the upward direction and the waggle run direction, and the distance from the hive to the food source is related to the duration of the waggle run [1]. Bees in close proximity to the dancing bee (followers) receive this information, and some of them may be recruited to the same food source, with many foragers ultimately visiting there [2].

A large number of studies have reported the efficacy of the dance for food collection in a controlled environment [3,4,5]. There are, however, only a few studies in which the dance behavior was evaluated under natural conditions. Sherman and Visscher [6] compared foraging success at natural food sources by measuring the mass of experimental colonies. They found that bees that were allowed to perform a dance with the directional information intact (an oriented dance) could generate more food collection than those who performed a dance in which the directional information was disrupted (a disoriented dance). Importantly, a significant difference was found only in winter and not in summer or autumn. Experiments in different habits within a year found that foraging efficiency was substantially impaired in a tropical forest, but was not significantly impaired in a temperate habitat when dance information was lost [7]. Recent experiments [8] performed in three cities in three different years found that physically preventing dance communication enhanced the loss of the colony weight in autumn (Figure 2A). Although those studies show that the fitness benefits of the dance are influenced by experimental area-, season-, year-, and colony-specific effects, those studies proposed that these dances help a bee colony to find food, particularly in habitats in which food is scarce.

A waggle dance generally consists of more than one pair of a waggle run and a return run. Therefore each of the waggle runs conveying directional information contains a certain range (±10–15°) of error from a mean direction of all waggle runs that a dancer performed [9,10,11]. The number of waggle runs in a dance increases as the profitability of an artificial feeder (e.g., concentration of sucrose solution) increases at both individual [12,13,14] and colony levels [15]. Consequently, the colony can rapidly respond to changes in the foraging environment. In Seeley’s experiment [15], two feeders were placed 400 m away from the hive, one to the south and one to the north. The concentrations of sucrose solution in the south and north feeders were 2.5 M and 1.0 M, respectively. After 4 h, the concentrations were changed to 0.75 M and 2.5 M, respectively. They found that the colony altered its feeder visitation habits in response to the change in the concentration of the sucrose solution. Mathematical modeling suggests that such adaptive colony-level decision making can be achieved not by a central command system but on an individual basis [16] and that the dance is beneficial under such a dynamically changing environment (Figure 2B) [17,18]. Furthermore, such flexible foraging may be achieved by an error in dance information. Simulation experiments [18] showed that flexibility of foraging depends on the degree of error. When the error range is large, such as 30° or more, the dance was not beneficial for food collection. When the error range is small, such as 0° or 5°, the successful rate of foraging was enormously high but in many cases failed to cause a switch to visiting the best feeder after the foraging environment changed. In the case of a natural error range such as 10–15°, the dance was beneficial and the bee colony was able to successfully visit a new feeder when the environment changed.

Recent advances in the understanding of the dance shine a light on the importance of odors during dancing. Successful foragers may encounter flower odors during nectar collection. If a dancing bee has an odor that the follower experienced previously during her successful foraging, this follower will fly to her previous feeder even if the dance indicates another feeder [19]. Dancing bees emit substances during dancing that encourage bees to fly out from the hive [20]. These findings imply that the effect of dancing may have two different functions: Informative (transfer the food location) and motivational (increase the number of bees that fly out to forage) functions.

Although the outcomes of bee activities (e.g., food collection, rate of successful recruitments, and dance occurrence rate) vary widely among colonies, sometimes negative [21], it is well established that a colony can achieve an effective collection of food via the dance [22].

## 2. The Follower’s Behaviors Induced by Various Waggle Dance Signals and How These Signals Are Sensed

*Apis florea*, considered the most primitive in *Apis*, has a habit to build an open-nest, not like *A. mellifera*, which suggests this species could use visual signals for their communication including the waggle dance. Other Apis species, *A. mellifera* and *A. cerana* live in lightless hives, which suggests these species must use other than visual sensory signals. Honeybees possess olfactory communication for sharing floral scents. Before and after the waggle dance, the dancers often contact their mouthparts with their hive mates via trophallaxes [23]. This trophallaxis behavior is useful for transferring not only the nectar but also a nectar-associated floral odor. During the trophallaxes, the receiver bees can learn the floral odor. The receivers are subsequently attracted to the waggle dance through this olfactory memory. Recently it was shown that the antennal contacts alone can act as a reward stimulus in the olfactory proboscis extension response (PER) learning [24], suggesting the followers might learn some aspects of the waggle dance via such rewarding contacts. The dancer also emits a non-floral scent during the waggle dance. The waggle dancer produces and releases two alkanes and two alkenes [20]. These odors increase the number of foraging bees, suggesting that these scents have another potential cue not only for recruiting foragers but also for drawing attention to the waggle dancer. On the honeybee antennae there are three morphologically identifiable putative olfactory sensilla: The sensilla placodea, sensilla trichodea type A, and sensilla coeloconica [25]. In the primary olfactory center, the antennal lobe, the spatial odor map has been established by Ca^2+^ imaging [26] and associative odor learning modifies neural representations of the odor map [27]. Unfortunately, we still do not know how the learned odor information increases both the foraging and attraction to the waggle dancer.

As mentioned above, the honeybees are attracted toward the learned odor, as it is associated with a reward. In the hive it is difficult to analyze the odor-induced behavior because it is often disturbed by congestion. Therefore, in our study, tethered insects with a fixed dorsal tergum on a floating ball were used to analyze the behavior of individuals free of such disturbance. We analyzed the locomotion patterns induced by reward-associated odor on proboscis extension response (PER) conditioning. In response to the learned odor, bees walked locally with alternate left and right turns during odor stimulation to search for the reward-associated odor source. Just after the learned odor stimulation, bees walked long paths with large turn angles to explore the odor plume [28]. It suggests that the bees search around the learned odor source by alternating left and right turn walking, and when the bees lose the odor plume they explore the odor in a long path with large turn angles.

The waggle dance produces two types of vibration: First, at low frequency (15–25 Hz) through abdominal movements from one side to the other side and second, at high frequency (250 Hz) through wingbeats. These vibrations can be transmitted on the surface of the “dance floor” [29]. The forager bees dancing on open, empty combs recruit three times as many nestmates to feeding sites as those that dance on capped brood combs, because of the smaller resonance, suggesting the vibrations propagate better and therefore have higher amplitudes on an open, empty comb [30]. On the other hand, it was suggested the vibrations were too weak and probably unreliable as a source of specific information about the velocity and direction of the dancer during the waggle run [31]. Furthermore, anecdotal data suggests dancing robots not touching the wax comb surface could recruit followers successfully [32]. Vibrations on the comb seems to serve to attract unemployed foragers to the dancers, but not to include the specific information about the location of the food source.

### 2.1. Distance

Since von Frisch’s discovery, it has not yet been clarified which sensory signals in the waggle dance are critical parameters for encoding the distance to the profitable flower. Frisch and Lindauer [1] analyzed various parameters related to the movements, and finally suggested the duration of waggle run as the index of distance. A mechanical model mimicking the waggle dance movements and producing similar olfactory and mechanical signals revealed that both wagging movements and sound during the waggle run are critical for recruiting followers via the waggle dance [32,33]. The following study suggested that the waggle dancer produces an inaudible, pulsed sound while on the waggle run [34]. The waggle dancer typically produces near-field sounds during the waggle run. Honeybees may be able to detect this particle velocity sound, but not the pressure component of sound [35] and responded to particle velocity vibrations of less than 500 Hz [36]. The rate of pulse vibration is rather consistent with the distance to the dancer’s indicated flower [37]. The pulses are produced by dorso-ventral wingbeats with one to five beats separated by intervals of motionless wings [38]. The pulse rate is also constant, independent of the food profitability [34]. Since the number of pulses increases linearly with the duration of waggle run, one possible parameter in the near-field sound for encoding the distance to the flower was thought to be the number of pulses in the waggle run [31,36], in addition to the duration of the waggle run (WD) (Figure 3) [36]. 

Near-field sound is detected by the vibration-sensitive sensory organs in the antennae, called Johnston’s organs (JO) [39]. It has been suggested that the JO plays an important role in detecting particle velocity vibrations caused by dance. For example, graded ablation experiments of the antenna revealed that the follower bee with one amputated antenna (including JO) was not recruited to an artificial feeder [40]. Although an unspecific effect due to the damage made to the antennae could not be excluded, both antennae have to be used for the follower (receiver) to obtain the information from the waggle dance. Additionally, the mechanical characteristics of matured honeybee antennae and the response properties of JO neurons are best tuned to detect 250–300 Hz sound generated during waggle run [41]. The JO seems not to detect vibrations with low frequency caused by wagging movements, however may extract the low frequency vibration through the central processing (see the next section). 

The central projection of JO has been identified [42,43]. The sensory afferents are divided into three antennal mechanosensory centers, the dorsal lobe (DL), the medial posterior protocerebral lobe (mPPL), and the dorsal gnathal ganglion (dGNG). The axon terminals only in the mPPL show some degree of somatotopy, but this is not the case in the other neuropiles, DL and dGNG. This suggests that there is a functional difference in the processing between mPPL and the other neuropiles. In *Drosophila*, the antennal mechanosensory center including the arborization of the vibration-sensitive interneurons has been identified and is referred to as the antennal mechanosensory and motor center (AMMC). The AMMC has five zones, A–E, which have somatotopy [44]; nevertheless, it is still difficult to discuss the corresponding regions in the honeybee.

Honeybees, like other insects, accumulate electric charge in flight, and when their body parts are moved or rubbed together. Greggers et al. [45] suggested the electric fields emitted by dancing bees induce passive antennal movements in stationary bees. They recorded the neuron response from JO to electric field stimuli. The electric fields emanating from the surface charge of bees might function as stimuli for mechanoreceptors to detect the particle velocity vibrations caused by the waggle dance.

### 2.2. Direction

The dancer indicates the azimuth to a target by their body orientation in the waggle run. How does the follower detect the dancer’s body orientation? Most followers face the dancer laterally and extend their antennae towards her body to get in direct contact [46]. In the study, it was found that the higher the number of the dancer’s wagging movements, the higher the number of the follower’s antennal deflections and that the time pattern of the follower’s antennal deflections depend on the angular position of the follower to the dancer during the waggle run. It suggests that the followers could detect the dancer’s body orientation by the time pattern. Gil and De Marco [46] suggested that mechanosensory input, presumably processed by neurons of the antennal joint hair sensilla, contributes to detect the time pattern of the antennal deflections for estimating the dancer’s body orientation in the waggle run.

## 3. Neural Processing of Distance Information Encoded in Waggle Dance

The duration of the waggle run increases proportionally with the distance to the flower source [1]. During the duration of the waggle run, the dancers vigorously shake their abdomens with low frequency at 15–25 Hz, while beating their wings at about 265 Hz. The other individuals follow the dancer’s abdomen, receiving low frequency vibration pulses composed of the dancer’s wing beats with high frequency. How could the follower bees encode the highly complex vibrations? The pulses have a constant pulse duration (PD) of around 16 ms and a pulse period (PP) of around 33 ms (Figure 3A). Anatomical and physiological evidence suggests a neural circuit for processing vibration pulses in the honeybee brain [47]. An identified DL neuron, DL-Int-1, is a GABAergic inhibitory neuron (Figure 3B). DL-Int-1 shows spontaneous activity, but when trains of pulses with short pulse period (short PPs) are applied to the antenna, DL-Int-1 shows remarkable hyperpolarization and the spontaneous spikes disappear (Figure 3C, left column). A post-inhibitory rebound (PIR) excitation (Figure 3C, arrowheads) appeared upon the offset of the pulse train. Under such stimulus conditions, DL-Int-2, a presumed postsynaptic neuron of DL-Int-1, evokes continuous spikes (Figure 3C, top left). In contrast, when trains of pulses with long PPs are applied to the antenna, DL-Int-1 shows intermittent spikes during the train of pulses (Figure 3C, right column, asterisks), and DL-Int-2 often shows a lack of spikes with remarkable inhibitory postsynaptic potentials (IPSPs) (Figure 3C, dots). These data suggest that the honeybee may use a disinhibitory network to encode the WD: DL-Int-2 spiking upon excitatory input from JO afferents is elicited by an inhibition of the presynaptic inhibitory neuron DL-Int-1 (Figure 3D). DL-Int-2 spikes in response to stimulation by trains of pulses with short PPs (Figure 3C, left), presumably as a result of the short-PP selectivity of the inhibition from DL-Int-1. Therefore, the disinhibitory network contributes to the coding of not just the WD, but also the short PP. These experimental results suggest a motif that resembles the functions of a “stopwatch” [48].

In addition to the particle velocity vibrations caused by wingbeats, waggling movements both cause tactile contact of a follower’s antenna with the dancer’s body and modulate the electrostatic field. They may also function as a signal related to the waggle dance and pick up the low frequency component of abdomen wagging [3,31,46]. The tactile contact and the modulated electrostatic field deflect the antenna, which may be detected by neurons in the antennal joint hair sensilla. These sensory afferents also project to the dorsal lobe [42,43], implying that the identified dorsal lobe interneurons discussed here might also be involved in the processing of the low frequency component of abdomen wagging.

## 4. The Evaluation of the Neural Circuits for Processing Distance Information Encoded in the Waggle Dance Using the Honeybee Standard Brain (HSB)

All the studies focusing on the potential neural substrates of distance coding have been limited to the pulses of high frequency airborne signals as received by the JO. Although the duration of the waggle run correlates well with the distance flown by the dancer, the other components of the waggle run correlate also with the distance, the number of waggles, the length of the waggle run and the duration of the whole round including waggle run and return run. Furthermore, the high frequency components of airborne signals from the vibration of the wings are just only one of several stimuli produced by the waggling dancer. It will be important for future work to include all these signals and study the whole range of frequency components. The picture emerging from the data collected so far allows to propose a first glance into the potential neural circuitry of distance coding.

A fundamental approach to the analysis of local neural circuits with a specific function is the measurements of neural response to a stimulus related to the behavior. For analysis of a local circuit for waggle dance information processing, neural responses in the DL, were recorded to a waggle dance-mimicking vibration stimulus on the antenna [47]. The recorded neuron was stained through the injection of a fluorescent dye, and it was then imaged by confocal microscope. In order to clarify the synaptic connection between neurons, simultaneous recording of multiple neurons is required, but this is not easy to accomplish. Due to the constraints of the experiment, one or only a few neurons are visualized in one brain sample using an intracellular staining technique. Even when using other staining techniques, only a limited number of neurons are visible in one brain. It is therefore necessary to integrate structural information taken from different brain samples for analyzing the connectivity between neurons. With such experimental difficulties, computational approaches based on the experimental results could be useful for analyzing the local neural circuit.

Morphological modeling and morphometric analysis of neurons could represent a useful approach for analyzing the neural circuit. Neurons have a complex structure similar to tree branches, and they have own characteristics in their arborization patterns. Software for automatic extractions of morphology from confocal and electron microscopy images are being developed and applied to various neural segmentations, but trial and error approach is still needed to extract complex neural structure. For example, we are currently extracting morphologies and constructing the model of neurons, which are arborizing in the DL region, by combination of automatic and manual segmentations [49]. Currently, morphological models have been generated for various neurons and shared in an online database [50,51,52,53]. In many morphological modeling projects for neurons, the structure of the neuron is described and shared in the SWC format [54]. Model neuron structures in SWC format can be visualized and used with imaging data by various software packages, such as Vaa3D [55] and neuTube [56].

The standard brain of the honeybee, HSB, is used as the platform for integrating morphological information of neurons [57]. The registration, which is transformed neuron morphology fitted into the standard brain or reference brain image, is an important process. The rigid and non-rigid image transforms are applied to register the brain region and neuron into the standard brain. Various image processing software and libraries, such as Fiji [58] and Insight Toolkit (ITK) [59], are providing functions for registration. Registered neuron morphologies provide us with the information on the connection among neurons (Figure 4). For an example, by measuring the distance among the branch terminals of registered neurons, it is possible to estimate the synaptic strength of connections between two neurons because a close distance between the terminals of the branches is considered to be a necessary condition for forming a synapse.

At present, the estimation of synaptic strength is generally analyzed based on the positional relationship between axons and dendrites of neurons based on light microscope images. However, further experimental studies would be necessary for determining synaptic strength more accurately. A detailed analysis based on the electron microscope image would be suitable for analyzing details of the synaptic connection morphologically. However, questions still remain, for example, about what the functional or actual strength is. Simultaneous measurement of multiple neuron responses by multiple electrodes or optical recording method might provide us with the dynamical properties of neural connections [60]. By applying these experimental methods on the brain region related to waggle dance processing, it can be expected that mathematical models showing the actual neural circuit structure and function could be reconstructed.

## 5. Computational Analyses of the Neural Circuits Processing the Distance Information Encoded in the Waggle Dance

While waggle dance communication of honeybees has been studied at the behavioral level for decades, investigation of the underlying neural processes has only recently begun [41,42,43], and open questions remain even at the early stages of processing of the waggle dance vibration signals [47]. Particular insights however have been gained from advanced computational analyses and modeling studies.

Interneurons DL-Int-1 and DL-Int-2 in the antennal mechanosensory center of the honeybee respond to stimulation of the antennae with artificial vibration pulses similar to those elicited by dancer bees during the waggle dance (see above). Properties and timing of the responses of these neurons suggest a disinhibitory circuit involved in the processing of waggle dance vibration pulses (Figure 3D). Disinhibition and PIR have been implicated in achieving high temporal precision in sensory encoding, not only in insects (for review see [48]) but also in vertebrates, on various time scales (e.g., [61,62]). It seems feasible that these mechanisms might also play a role in representing the temporal signals of the honeybee waggle dance. To clarify the plausibility of the proposed circuitry, model simulations have been performed [63]. The putative circuit with input from JO and inhibition between DL-Int-1 and DL-Int-2 (Figure 3) was simulated with model neurons (http://modeldb.yale.edu/239413). When stimulation with vibration stimuli of different pulse parameters was simulated, the model neurons showed responses that qualitatively resembled the corresponding experimental data (Figure 5A). The inhibitory connection from DL-Int-1 was essential for the dependence of DL-Int-2 responses on stimulation pulse parameters in the model [64]. Thus, disinhibition is likely to play a role in encoding the timing of waggle dance vibration pulses in the follower honeybees. Disinhibition might be efficient in achieving precisely timed responses by separating the timing mechanism from the response generation.

Evidence for the relevance of the interneuron circuits in the honeybee antennal mechanosensory center for the waggle dance communication comes from investigations of changes in these neurons during maturation [64]. Adult honeybees spend the first days after emerging within the hive with cleaning and nursing tasks and do not take part in foraging. Only after several days do they begin to leave the hive for foraging trips and to participate as followers in the waggle dance [65].

Kumaraswamy et al. [64] analyzed the morphologies of DL-Int-1 from young honeybees that were just emerged and compared them to neurons from older forager honeybees. A novel alignment method was used to achieve spatial registration of neuron morphologies without the need of landmarks or other information [66]. This enabled the calculation of quantitative, spatially resolved statistics of the branching properties of the neurons of bees of different age classes. While the overall morphological structure was found not to differ between neurons of young and forager bees, the analysis revealed gradual, localized changes in dendritic density (Figure 5B) in specific regions of the arborization that could be associated with input and output regions of the neuron [64]. This suggests that, while the coarse neural circuitry for vibration processing is already established when the bee emerges, a refinement process is ongoing that further adapts the neurons to the sensory demands as the bee takes on new tasks.

Consistent with the specific changes observed in morphology, the responses of DL-Int-1 neurons showed changes indicating specific adaptations in signal processing. While the main features and overall pattern of response (i.e., on-phasic excitation, tonic inhibition, and post-inhibitory rebound) were preserved across age groups, there were quantitative changes between young and forager bees, notably an enhanced difference between baseline and inhibitory response, and increased response amplitudes in on-phasic response as well as post-inhibitory rebound [64]. These changes consistently indicate that the circuitry that signals vibration pulse patterns achieves a higher signal-to-noise ratio in forager bees as compared to newly emerged bees.

While the question remains open whether the observed differences between young and older honeybees are a result of a predetermined maturation process or of sensory experience, these findings are in line with patterns of development in other structures [67,68] and indicate a refinement of connectivity and cellular response properties for reliable encoding of the information conveyed in the waggle dance.

## 6. The Mechanism for Detecting the Azimuth and Distance toward the Feeding Site

The honeybee recruits that decipher the spatial information encoded in the waggle dance start their flights at the hive entrance. During the foraging flight the foragers orientate themselves to the indicated direction. As a result of sunlight scattering in the atmosphere, the skylight is partially plane-polarized, and the celestial e-vectors (electric-field vectors of the light waves) are arranged in a concentric pattern around the sun [69,70]. It is well known that many insects, including honeybees, exploit this skylight polarization to deduce orientation (for review see [71,72]). When an artificially polarized light is presented to a dancer performing its dance on a horizontal comb, the dance orientations are significantly shifted depending on the e-vector orientation of the light [1,73]. Under unpolarized light, the forager shows directionally random dances (i.e., it totally fails to transfer the directional information to the food source) [6]. Moreover, it has been reported that the dance directions are also modulated by the e-vector orientations of the light that the animal experienced during the foraging trip [74]. These facts clearly indicate that the honeybees utilize polarized skylight as an orientational cue for dance behavior. In contrast to many such investigations regarding the waggle dance orientation, only a few studies have been conducted to examine behavioral responses of a flying bee to celestial e-vector information. Kraft et al. [75] showed that bees trained to fly in a four-armed maze to a feeder, in which the bees received polarized light stimulus from above, chose their foraging routes as they received e-vector information, similar to what they experienced during the training. This suggests that the bees actually sense the e-vector orientation from the sky during the flight and use it for navigation.

Polarization vision in insects is mediated by a specialized region in the compound eye called the dorsal rim area (DRA), in which the ommatidia have extremely high polarization sensitivity because of their structural and physiological properties. In honeybees, the rhabdom is non-twisted and contains orthogonally arranged microvilli [76,77], which act as e-vector analyzers. The optical property of the ommatidia is also different from that of other eye regions. The cornea of the DRA contains light-scattering pore canals [78], such that the visual field of the photoreceptors could be increased. The photoreceptors in DRA are known to be UV-sensitive [79], and therefore a UV range of light is necessary for the honeybees to detect the celestial e-vector. Central neural processing for e-vector information in the brain has been investigated in various insect species such as locusts [80,81,82], crickets [83,84], dung beetles [85], monarch butterflies [86], and nocturnal bees [87]. In these insects, various types of polarization-sensitive neurons are found in the central complex (CX) and the CX is strongly suggested to derive compass information from zenithal polarized light input. In the honeybee brain, as far as we know, no polarization-sensitive neurons have yet been identified. However, the possible neural pathway from DRA to CX has been anatomically revealed [88,89]. Considering that the morphological and physiological properties of the polarization-sensitive neurons in the CX are highly conserved among species, a similar compass system likely exists in the honeybee brain.

Skylight polarization is a line symmetrical pattern in which the solar meridian is the axis. Therefore, e-vector information from the sky indicates the orientation of the animal’s body axis relative to the solar meridian. However, the e-vector alone is not sufficient to deduce the heading azimuth (i.e., the bee cannot discriminate certain azimuth φ from the azimuth φ + 180°). How does the bee uniquely identify its heading azimuth? It has been reported that the bee discriminates between the solar and the antisolar half of the sky based on the spectral cue of the skylight [90]. In addition to the polarization pattern, sunlight scattering in the atmosphere also produces spectral gradients along the sky because of the difference in scattering efficiency based on the light wavelength. It causes a different spectral contrast between the solar and the antisolar half: The light of long wavelength dominates in the solar half while a relatively smaller content of long wavelength light does so in the antisolar half [91]. When dancing foragers are given a patch of unpolarized long-wavelength (green) light, they interpret it as sun. In contrast, a short-wavelength (UV) light is expected to exist somewhere within the antisolar half of the sky [90]. Integration of polarization and spectral information of the skylight may help the bee find a correct heading azimuth. In locusts, it has been suggested that the anterior optic tubercle, which sends polarized light information to the CX via the lateral accessory lobe, may contribute to such an integration [92]. 

Behaviorally, it has been proven that honeybees estimated their travel distances by optic flow, an image flow caused by self-motion. Foragers flying through short, narrow tunnels with visually textured walls perform waggle dances that indicate an overestimated flight distance [93,94]. Behavioral tests with the optic flow stimulus of various color and contrast combinations have revealed that the motion detection in honeybees is colorblind and that the green receptor signal is used for distance estimation [95]. The neural mechanism underlying such an odometer based on the optic flow is still largely unknown. However, in a nocturnal bee species, a group of neurons in the noduli, one of the input sites of the CX, was found to encode the direction and speed of the optic flow stimulus, indicating that they can potentially convey travel distance information to the CX [87]. Because, as mentioned above, the CX is considered to be an internal compass, it has been strongly suggested that, in the CX, the information on azimuth and distance towards the feeding site are integrated during the foraging trip and the computational process for path integration is conducted [96], which is necessary not only for returning to the nest but also dancing in the nest [97]. It is still unknown whether the CX is the premotor center executing the waggle dance in the brain and the manner in which the premotor center controls the waggle dance movements, but the encoded azimuth and distance information in the CX during the foraging trip must be decoded as two parameters: The orientation and the duration of the waggle run. Recent studies suggested the majority of bees needed two or more foraging trips to update dance duration and also showed intermediate dance durations during the update process, which implied that it takes several processing steps to update the information for transposing flight navigation information to the dance behavior [98]. Moreover the dance follower could orientate by using the distance and azimuth information learned from the waggle dancer. It is also interesting how the foraging flight could be navigated by the spatial information decoded from the waggle dance. 

## 7. Conclusions

In this review we summarized the current research related to waggle dance communication and its potential neural mechanisms. We are still far from understanding the complete neural mechanisms for encoding and processing the waggle dance information, but our neuroethological approach combined with computational analyses using the HSB has been fruitful in the past and offers a promising path for future progress.

## Figures and Tables

**Figure 1 insects-10-00336-f001:**
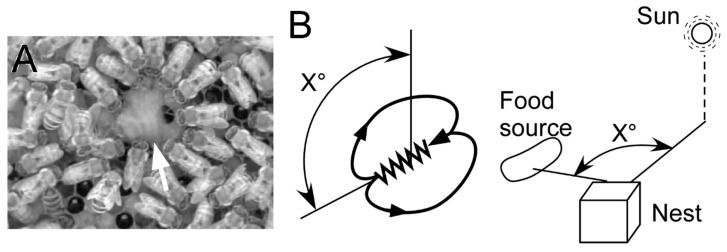
The waggle dance. (**A**) A dancer (arrow) on the waggle run and many bees (followers) around the dancer. (**B**) Encoding a direction of a food source. The direction of a waggle run in relation to the position of the sun when dancing on a horizontal comb and in relation to upwards/anti-gravity when dancing on a vertical comb indicates/represents the direction of a food source.

**Figure 2 insects-10-00336-f002:**
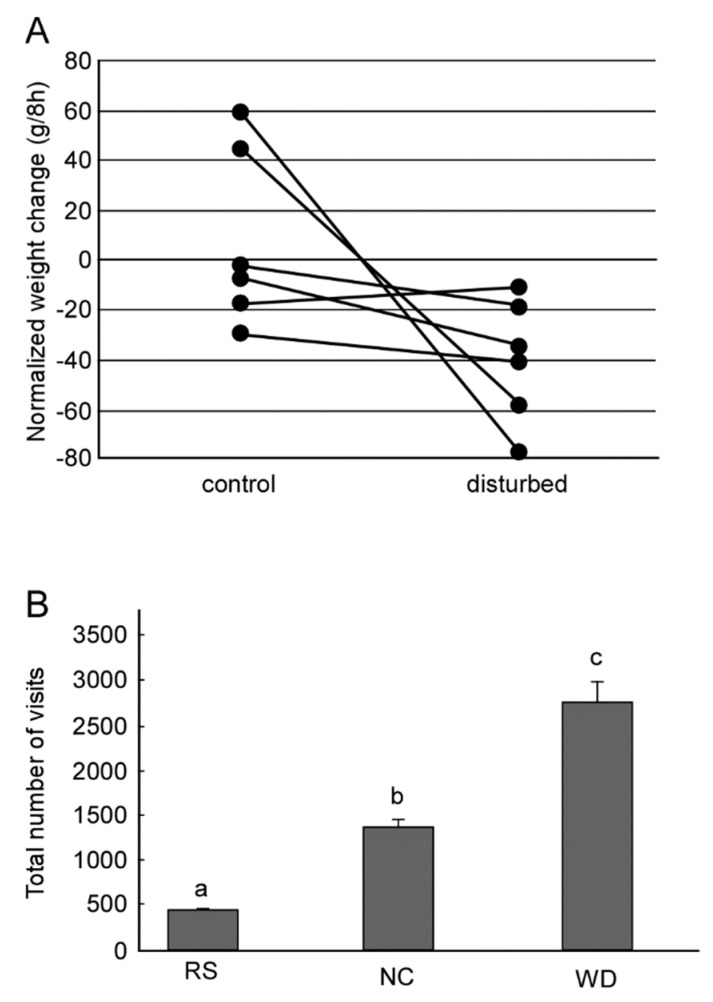
Availability of dance for food collection. (**A**) The daily weight changes. Six colonies were examined. One colony measured its weight in both dance-preventing (disturbed) and non-preventing (control) conditions. Dots connected by each line represent each colony. (**B**) Total number of visits to feeders of three virtual colonies in terms of visits to feeders. Foraging characteristics is different among the virtual colonies. Bees in random-search colony (RS) make foraging without locational memory of food sources or dance communication. Bees in no-communication colony (NC) memorize the location of the food source if they found and use it for later foraging, but make no dance communication. Bees in a waggle dance colony (WD) memorize the location of the food source and use it, and perform waggle dance to transfer locational information. The letters above the boxes represent groups showing statistically significant differences (ANOVA and post-hoc Tukey–Kramer test). (A,B were modified from [8,18], respectively).

**Figure 3 insects-10-00336-f003:**
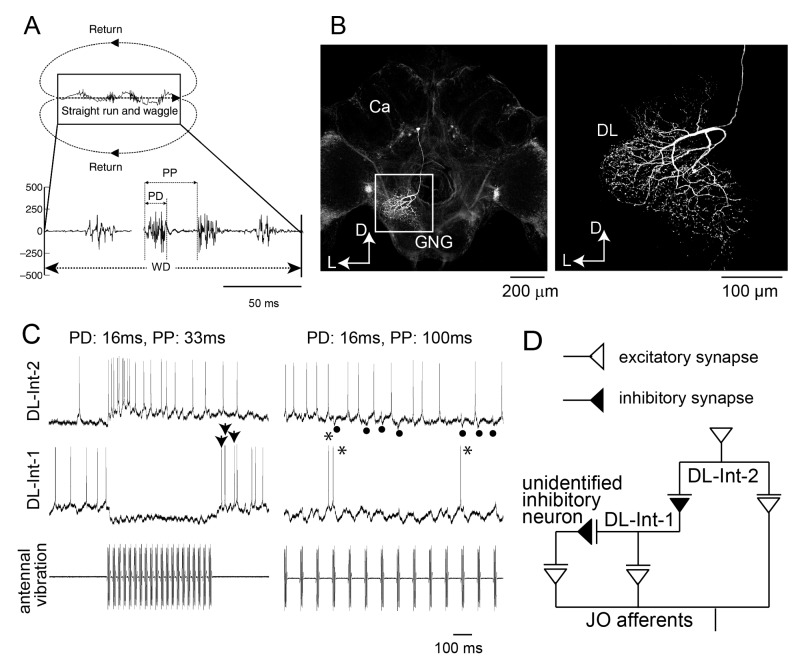
(**A**) (Top) Movement trajectory of a honeybee during the waggle dance. The dance consists of a waggle run and a return run. The distance to the flower source is encoded as the duration of the waggle run (WD) of the dance. (Bottom) Thoracic vibration velocities recorded during the waggle run. Intermittent vibration pulses occur with a constant pulse duration (PD) of about 16 ms and a pulse phase (PP) of about 33 ms. (**B**) Morphology of DL-Int-1. (Left) Gross image of the DL-Int-1. (Right) Magnified image of arborization in the dorsal lobe (DL) indicated by a square region in the left image. Ca: Calyx of mushroom body; GNG: Gnathal ganglion; D: Dorsal; L: Left. (**C**) Intracellular records of dorsal lobe interneurons 1 (DL-Int-1, middle) and 2 (DL-Int-2, top) in the antennal mechanosensory center of the honeybee in response to vibratory mechanical stimulation to an antenna (bottom). Left: When the PPs are shorter than 50 ms, the DL-Int-1 receives strong inhibition that allows no spikes during the pulse trains and exhibits a post-inhibitory rebound (PIR) excitation (arrowheads) upon the offset of the pulse train. DL-Int-2 exhibits elevated spiking activity during stimulation. Right: DL-Int-1 shows spikes (asterisks) intermittently during the inter-pulse interval phase when the PP of the stimulus is longer than 50 ms. Under these conditions, the DL-Int-2 often shows a lack of spikes with remarkable inhibitory postsynaptic potentials (IPSPs) (dots). (**D**) Putative neural circuits for processing the temporal structure of vibration signals. JO: Johnston’s organ. Modified from [34] with the permission of Company of Biologists for A, and from [48].

**Figure 4 insects-10-00336-f004:**
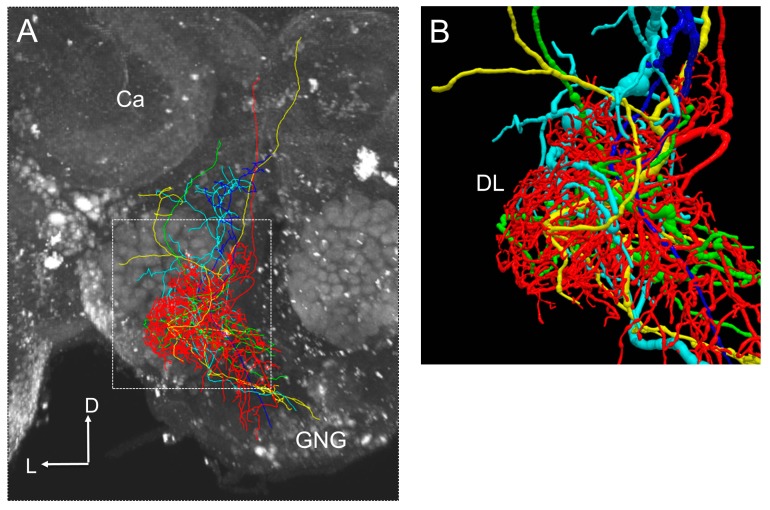
(**A**) An example of interneurons shown a response to a vibration stimulus presented to the antenna. Each interneuron is shown in a different color. (**B**) Zoom of arborization area of interneurons in the antennal mechanosensory center of the honeybee brain. Presumable synaptic connections could be estimated from locations of neurite terminals.

**Figure 5 insects-10-00336-f005:**
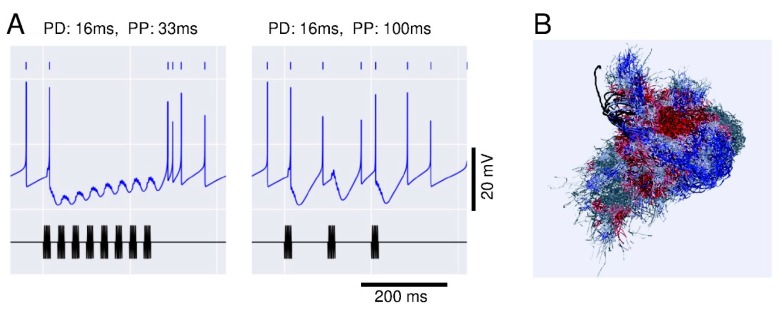
(**A**) Responses of model DL-Int-1 neurons in network simulations of the disinhibitory circuit (see Figure 3D) to antennal vibration pulse trains. Black traces: Antennal stimulus. Blue traces: Model membrane potential. Spike times are in addition indicated by blue vertical lines. Modified from Kumaraswamy et al. [63]. (**B**) Region-dependent changes in dendritic density between young and forager bees. Co-aligned morphologies of 12 DL-Int-1 neurons. Color indicates regions of decrease (red) and increase (blue) of local average dendritic density in neurons of foragers relative to neurons of newly emerged adults. Modified from [64].

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
