# Peer review of "Neuroethology of the Waggle Dance: How Followers Interact with the Waggle Dancer and Detect Spatial Information"

_insects, 2019, doi:10.3390/insects10100336_

Round 1
Reviewer 1 Report
Dance followers
In their review, the authors provide a detailed account of the honey bee waggle dance, with a focus on how distance and direction may be encoded in the dance and decoded by dance followers. I like the emphasis on the recent neuroethology of dance communication and think that this will be of interest to readers. My primary comments focus on the section describing sound communication.
Sound is defined as molecular disturbance propagating through a medium. Sound as relevant to bees is classically divided into two categories: substrate-borne vibrations and air-borne sounds. Airborne sound consists of two components: particle velocity (near-field) and pressure components. Bees cannot hear the pressure component, as the authors correctly point out. However, at several points throughout the paper, the authors refer to airborne vibrations, which I think is less clear than using the term “particle velocity” sound. “Particle velocity” more accurately conveys what they authors are interested in, particularly in how bee antennae and Johnston’s organs detect airborne sounds. I suggest that this be changed throughout.
My second suggestion is that the authors reference and mention the recent paper by Chole et al. (2019) since it is relevant to the question of odor learning from dancing honey bees and the importance of antennal contacts. The fact that antennal contact alone can act as a reward stimulus in olfactory PER learning is fascinating and directly relevant to how dance followers may learn the waggle dance. Although it is speculative, it is quite that followers learn some aspects of the waggle dance via such rewarding antennal contacts.
Finally, it would be good to clarify at the beginning that this review largely focuses on Apis melliferaand that, while we expect similarities in the other honey bee species, there may be differences in how dance followers receive waggle dance information. For example, in open nesting species, dance followers have access to visual information about the dancer which they typically lack in cavity nesting species.
Reference
MLA |
Cholé, Hanna, et al. "Social contact acts as appetitive reinforcement and supports associative learning in honeybees." Current Biology 29.8 (2019): 1407-1413. |
Minor suggestions
Line 21. I suggest, “dance to reach the advertised destination”
Line 58. I suggest “Although those studies show that the fitness benefits of the dance are influenced”
Line 117 “in our study,”
Line 128 “vibrations”
Line 129 “or that the vibrations propagate better and therefore have higher amplitudes on open, empty comb”
Line 142 “The waggle dancer typically produces near-field sounds during the waggle run. Honey bees may be able to detect this particle velocity sound, but not the pressure component of sound [31] and responded to particle velocity vibrations of…”
Line 144. I think you mean the “number of pulses is correlated with the distance to the indicated flower”.
Line 151. “dance,”
Line 161. This sentence is unclear, please rewrite. Do you mean that they need antennal contact, but not the antennal tips?
Line 166 “is near-field sound.”
Line 167. “Near-field sound is detected….
Line 264. For example,
Line 376. …compass system likely exists
Line 392 …to the CX…
Line 400 …in a nocturnal bee species…
Line 410. Please be more specific here, what is the interesting challenge?
Line 494. Rohrseitz
Author Response
Please see the attachment.
In the attachment, you could find out "See Line xxx".
The xxx is the line number that we changed in the word file with tracked change.

Reviewer 2 Report
This is an enlightening review that covers important issues and references of this field. I recommend publication of this review with minor corrections.
I have an impression that the comprehensiveness and readability are different among sections. For example, the section 6 is well-written and organized, but the section 4 is a little difficult to read and understand. If possible, it would be helpful for readers to edit the manuscript thoroughly by a good English writer.
L55-L57: ref [8] seems not to correspond to Fig. 2A. L193 and Fig.3C: I couldn't find any arrowheads. L197 and Fig.3C: I couldn't find any dots.Author Response
Please see the attachment.
In the attachment, you could find out "See Line xxx".
The xxx is the line number that we changed in the word file with tracked change.

Reviewer 3 Report
Major
It is apparent that the text was written by five authors as a higher level of homogeneity and consistency in the writing style is lacking. Also, some parts of the text seem to have been written in a rush (mostly section 2). Although the general structure is logic and the scientific content is correct and represents the current state of the field, I would recommend going through the text more thoroughly to increase readability and comprehensibility.
Minor
Line 13 The first sentence is not really informative and therefore dispensable, unless you want to refer to the Journal titel
Line 16 fascinating remarkable?
line 30 consider “communicate to” instead of “with” as there is no evidence so far that the recruits reply
line 38 consider two sentences (stop after “waggle run direction”) or use whereas instead of comma
caption Fig. 1 The direction of a waggle run in relation to the position of the sun when dancing on a horizontal comb and in relation to upwards/anti-gravity when dancing on a vertical comb indicates/represents the direction of a food source...
line 79 when listing arguments for and against the adaptive value of the waggle dance, you could also mention: R. I’Anson Price, N. Dulex, N. Vial, C. Vincent and C. Grüter Science Advances 2019:Vol. 5, no. 2, DOI: 10.1126/sciadv.aat0450
generally, you should add more recent papers should, e.g.:
Barron and Plath. Journal of Experimental Biology (2017) 220, 4339-4346 doi:10.1242/jeb.142778
Chatterjee et al. Journal of Experimental Biology (2019) 222: jeb195099 doi: 10.1242/jeb.195099
George, E.A. & Brockmann, A. Behav Ecol Sociobiol (2019) 73: 41. doi: 10.1007/s00265-019-2649-0
caption Fig. 2B unclear what random-search colony means, also not explanatory from the text
Line 98 Section heading should be phrased clearer, e.g.: The follower’s behaviors induced by various waggle dance signals produced by the waggle dance and how these signals are sensed
Line 104/105 The dancer also emits a non-floral scent…
Line 122 …produces two types of vibration(?): First, at low frequency (12-25 Hz) through body abdominal movements from one side to the other side and second, at high frequency (250 Hz) through wingbeats.
Line 123 These vibrations…
Line 125 capped brood combs, because of the smaller resonance [27].
Line 126 …it was suggested the vibrations were too weak and…
Line 128 …the (you introduce them here for the first time!) dancing robots…could recruited the followers…
Line 129 Vibrations in on the comb seems to serve to attract … about the location of the food source.
Line 125-130 You exclude the floor vibration as a location signal (and being only a motivational signal) based on Landgraf et al. 2018. In this unpublished (arXiv, non-peer-reviewed) paper, they report that only 2 (out of 6 labelled) bees followed somehow the robot’s dance to the feeder and the authors themselves state that their findings were only “anecdotal in numbers” (line 337). You should rephrase the argument more cautious, e.g. “anecdotal data suggests…”
Line 136 Previously, you were writing about vibrations and now you introduce sound. You should describe the difference.
Line 137-138 … that the waggle dancer produces (additionally to the low and high frequency described above) a third (?) vibrational(?) signal, which is an inaudible (for whom, humans or bees?), pulsed sound during the waggle run [citation missing]
Or is this sound produced by the wing beats and therefore congruent with the high frequency signal? Please make clear.
Line 145 Fig 3C is described before Fig. 3A
Line 147, 149 suggested that the JO
Line 151 both antennae could have to (?) be used (intact?)…
Line 158 consider updated nomenclature: dGNG instead of dSEG according to Ito et al. Neuron. 2014;81(4):755-65. doi: 10.1016/j.neuron.2013.12.017.
Line 166 you could structure the text more efficiently by using subheadings: line 131-145: distance
line 166-180: direction
Line 166 … in the waggle run
Line 167 This section seems to be written in rush and needs re-writing: Most followers face the dancer laterally and extend their antennae towards her body to get in direct contact.
Line 177 …electric fields were emitted by dancing bees induce…bees, and recorded…
Maybe consider alsotwo sentences.
Line 180 …caused by the waggle dance
Line 180 “The electric fields … of bees (might?) function as … caused by the waggle dance.“
(As you might know) Uwe Greggers suggests the direction to be encoded in the “pendulum's resonance frequency”. But it is still unpublished. Nevertheless, you could refer to their website: (http://www.honeybee.neurobiologie.fu-berlin.de/column/pendulum%20theorem.html)
Line 184-186 I don’t understand the sentences: “The other individuals follow the dancer’s abdomen, receiving low frequency vibration pulses caused by the dancer’s wing beats with high frequency.” Why are low frequency vibration pulses caused by high frequency wing beats? Is it two sentences? Please make clear. Also, do you mean here the inaudible pulse sound, described in section 2? Do we have three types of vibration, 1. low f (12-25 Hz), high f (350 Hz) and pulse sound, shown in Fig. 3A, or is pulse and high f the same?
Line 184 15-25 Hz (low frequency) is incongruent with the other section (line 123) in which you wrote 12-25 Hz. Please be congruent.
Line 192/193 (Fig. 3C, arrowheads) arrowheads are missing
Line 197 (Fig. 3C, dots) there are no dots
In general: low frequency vibration is caused by body waggling and might encode distance
high frequency vibration is caused by wing beats and might encode direction
line 235 also with the distance:
line 256 additionally to ref [46] you could also refer to the insect brain database: insectbraindb.org
lines 278-282 What is meant by “optical recording” methods.? Ca2+ imaging or transgenic channelrhodopsin? Because the latter is still very futuristic in comparison to e.g. multiple electrodes and the paper you refer to is a Drosophila paper using all these fancy methods that are not applicable in bees. You could make that more slearly.
Caption Fig. 4 shown … response to a vibration stimulus presented to the antenna.
line 296 “antennae with artificial vibration pulses”
line 361-362 “These facts clearly indicate that the honeybees, like many other insects, utilize polarized skylight as an orientational cue for dance behavior.” It sounds like many other insects would do dance behavior. Consider deleting “like many other insects”.
Line 381 right: polarization-sensitive neurons in the CX have not been identified. But the possible anatomical pathway from the DRA to the CX has been revealed: Held et al. 2016; Front. Behav. Neurosci., 2016 https://doi.org/10.3389/fnbeh.2016.00186
Author Response

(The authors gave the same response as above.)

Round 2
Reviewer 1 Report
The authors have satisfactorily addressed all of my comments, and I think that their manuscript is now ready for acceptance.